# Site-specific growth dynamics and yield patterns in *Populus deltoides* against edaphic variability over seven age-gradations

Ashok Kumar Dhakad[1]*, Tarsem Singh[1], Avtar Singh[1], Kuldip Singh[2]

1 Department of Forestry and Natural Resources, Punjab Agricultural University, Ludhiana, India,
2 Department of Soil Science, Punjab Agricultural University, Ludhiana, India

* ashokdhakad@pau.edu

## Abstract

Most studies are focusing on general soil fertility effects on poplar growth rather than site-specific variations in edaphic factors considering the soil texture and profile as factors on poplar productivity and increments with age. Poplar (*Populus deltoides* Marsh.) grown in three different types of soil and across seven age-gradations were evaluated for growth behaviour and productivity pattern. We selected three nearby; even-aged poplar stands (PS1 with loamy sand, PS2 with silt loam, and PS3 with sandy loam texture) from poplar plantations, each with varying edaphic and similar agrometeorological conditions. We collected a composite soil sample for each transect at three distinct soil depths (0–30, 30–60, and 60–90 cm) and determined the physicochemical characteristics of the soil. Significant differences in soil texture and characteristics were noted across the three poplar stands (PS1, PS2 and PS3) due to variations in soil particle composition with depth. PS2 had more silt and clay with higher electrical conductivity (EC) and saturated hydraulic conductivity (SHC), while PS1 was richer in phosphorus, and PS3 had sandy textures and higher organic matter content. Early growth (height and diameter at breast height) was highest in PS3 but transitioned to PS2 after the fifth year. Leaf area, crown spread, timber weight, and total biomass also peaked in PS2 across age gradations. Current Annual Increment (CAI) values for tree growth variables increased initially but declined after the third age-gradation. For pulpwood, biomass productivity was 9.28% higher than that of PS2 and 29.15% higher than that of PS1, while PS2's productivity was 15.60% higher than that of PS3 and 45.81% higher than that of PS1 due to favorable texture and nutrient retention required for poplars. The study found the highest productivity (42.40 m³ha⁻¹yr⁻¹) in PS3 during a 5-year cycle, but PS2 excelled in subsequent years due to consistent nutrient availability. Thus, optimal growth was observed in soils with a balanced composition of sand (75–80%), silt (6–8%), and clay (13–17%). Soil organic carbon and phosphorus showed strong positive correlations with timber weight and growth parameters. The findings underscore the need for tailored

**Data availability statement:** Data is already included in the Manuscript itself.

**Funding:** The author(s) received no specific funding for this work.

**Competing interests:** Authors have no competing interests.

management practices, including soil preparation before planting followed by precise fertigation to optimize the productivity of poplar plantations.

## Introduction

Increasing demand and declining availability of wood has prompted a renewed interest in growing short rotation woody trees either in plantations or under agroforestry systems [1]. Plantation forests cover about 131 Mha and 44% of plantation forests are composed mainly of introduced species globally [2]. In India, area under agroforestry is 28.42Mha which contributes 8.65% to the total geographical area till 2024 [3]. The country has the potential to increase the area under agroforestry to about 17.57% of the TGA [4]. Globally poplar plantations cover 9.4 Mha which is more than double in last 15 years [5]. In India, the most preferred forest tree species in agroforestry practices is poplar especially in Northern India [3]. In Punjab, poplar species occupied an area of about 0.276 Mha (5.63% the state's area) with a reasonably good accuracy of 81% using high resolution multispectral remote sensing data (LISS-IV, spatial resolution-5.8 m) [6]. It is an interesting fact that farmers are getting approximately 46% more economic return from the poplar-based agroforestry practice than conventional rice-wheat land-use systems [7]. Thus, Poplar based agroforestry system and/or plantations are more gainful and helpful in doubling farmers' income over conventional agriculture practices in India [8].

The clone selection, type of soil, irrigation, fertilisation, climate, and management regime all have a significant impact on the productivity and biomass production of poplar plantations [9,10]. To maximize the growth of poplar, it is essential to understand the connections between growth rate, plant nutritional needs, and soil nutrient availability [11]. Although poplars can grow practically anywhere, they only achieve their maximum potential and function on well-suited sites [12]. Poplar grows excellent in soil having >10% soil porosity, clay content less 35% and with pH range of 6.5–8.0 [13]. Soil texture, physicochemical parameters and nutrient availability in surface soil are the important variables for successful establishment of poplar block plantations and to achieve its optimum growth potential [14]. The soil texture mainly affects the rooting behavior and its distribution pattern than above-ground growth with more root development if soil has sandy texture [15]. The performance of poplar depends on their complete ability to use most effectively nutrients and moisture on a particular productive site [16]. Nutrient content was highest in surface soil which decreased with increase in soil depth [17]; however, addition of litterfall from trees can improve the nutrient content of soil under trees and change the availability of these nutrients. There are a large number of research studies on poplar farming, but no systematic studies have analyzed the role of soil properties on growth of field plantation of poplar. In this connection, we hypothesized that i) site-specific variations in soil properties significantly influence the growth dynamics and yield of *Populus deltoides*, ii) the interaction between edaphic factors and tree age modulates the growth rate and biomass accumulation in *P. deltoides* plantations. Keeping in view, the present study

was planned with the following objectives: i) To assess the impact of site-specific soil variability on biomass accumulation and yield patterns in *P. deltoides*, ii) To analyze the relationship between edaphic factors and the growth dynamics of *P. deltoides* across seven different age-gradations.

## Materials and methods

### Study sites

The study sites were selected at farmers' field plantations planted in Khaira Bet, Nurpur Bet, Birmi and Basseimi villages in Ludhiana district, which is the core poplar growing area in northern India falls under the riverian area of Sutlej River (Table 1) in Indian Punjab at an altitude of 247 m amsl. The experimental sites are located in the north-western India falls under 6th Agro-Climatic zone, *i.e.,* Indo-Ganagetic Plains of India. The area is characterized with tropical to sub-tropical climate with a short period of wet season from July to September and long dry season from late September to June. Frost occurrence was not common. December and January were the coldest months, whereas; May and June were the hottest months. A ten-year average of temperatures from 2013 to 2022 showed that the range was 2°C in January to 46°C in May, with mean monthly maximum and minimum temperatures ranging from 17.4°C in January to 39°C in May and 6.0°C in January to 26.8°C in July, respectively [18]. The monsoon season, which runs from June to September, received the most rainfall (>75%) of the total 832 mm of precipitation annually (an average over ten years of measurements, 2013–2022). The Ludhiana district is made up of plain terrain; in which 99.8% of its agricultural land is irrigated, either by agroforestry or agriculture practices, and the water used for irrigation comes from high-quality tubewell or canal water. The soil was alluvial in nature and normally well drained falling in Inceptisols followed by Entisols [19].

Extensive surveys were carried out in Ludhiana district of Punjab, India to find the different soil texture classes (Table 1). The variability in the soil textural class is basically due to plantations lies in river basin belt of Sutlej River.The *Populus deltoides* clone L-47/88 was established in three stands (PS1, PS2 and PS3) in winter season, *i.e.,* January-February 2016 on well ploughed cultivable lands in these soil texture classes. The one-year-old saplings of poplar were transplanted in 90 cm depth holes which were dug by tractor operated earth auger, with spacing of 5×4 m (~500 trees ha-1). Similar irrigation rate and management regime for all poplar stands were applied as per the package and practice recommended by Department of Forestry and Natural Resources, Punjab Agricultural University, Ludhiana, India [20].

### Soil sampling and laboratory analysis

The physicochemical characteristics of each texture class with three soil depths were analyzed from the soil samples taken at the time of stand establishment stage (1st age-gradation during 2016) in the Department of Soil Science, PAU Ludhiana. In order to determine the physicochemical parameters of the soil in poplar stands, composite soil samples were routinely taken using a random sampling method from five places in each plot, excavating profiles to a depth of 0−30, 30−60, and 60−90 cm as the vertical pattern of root (> 5 mm root diameter) distribution is approximately 85 cm when poplar tree density is about 500 trees ha-1 [17,21]. Soil sample were dried on cemented floor at room temperature, and grounded thoroughly with the help of mortar and pestle to pass through 2 mm sieve [22]. These sieved samples were stored in polythene bag with proper labelling.

**Table 1. General information of three poplar stands.**

| Code | Place of planting | Latitude | Longitude | Soil texture class |
|------|-------------------|----------|-----------|--------------------|
| **PS1** | Khaira Bet village | 30°58'19.051''N | 75°40'74.759''E | Loamy sand |
| **PS2** | Nurpur Bet village | 30°57'54.947''N | 75°41'45.211''E | Silt loam |
| **PS3** | Birmi and Basseimi village | 30°55'37.284''N | 75°43'52.148''E | Sandy loam |

The physicochemical characteristics of each texture class with three soil depths were analysed under laboratory conditions. Soil texture was determined using the international pipette method [23]. Using a glass calomel electrode, the pH of the soil was measured in a suspension having 1:2 soil:water solution. After measuring the soil pH, the soil suspension was kept overnight for getting clear supernatant solution and after that Electrical Conductivity (EC; dsm$^{-1}$) was measured with an Elico conductivity meter (CM 180 model) using 1:2 soil-water suspensions after equilibrating the soil sample for one day [24]. The Saturated Hydraulic Conductivity (SHC; mm/sec) was directly estimated using texture data and organic matter content as per the method described by Nemes et al. [25]. The physicochemical properties of soil with varying depths are presented in Table 2. The Organic Carbon (OC; %) content in soil samples was estimated by wet combustion rapid titration method. Organic matter content was estimated by multiplying organic carbon percentage with the factor of 1.724 [26]. Alkaline potassium permanganate ($KMnO_4$) method was used to determine the available nitrogen [27]. The available phosphorus was determined by using the procedure given by Olsen et al. [28] and available potassium by using flame photometer [29].

### Growth and productivity measurements

At the end of the growing season, when the growth peak had already passed, the growth parameters of trees classified into seven age-gradations were measured from 2016 to 2022. For the purpose of measuring growth metrics, fifteen trees were randomly selected from each plantation in three replicates. Ravi's altimeter was used to measure the height of the tree from the ground to its apex, and a Vernier calliper was used to measure the diameter at breast height (dbh) at a height of 1.37 meters above the ground. Average leaf area ($cm^2$) was measured with CI-202 Portable Laser Leaf Area Meter by taking three mature leaves from the base, middle and upper canopy of the tree. The crown spread ($m^2$) was estimated with two poles holding straight and contacting the outmost tip of the opposite sides of tree. The CAI was calculated by differencing the current year growth with previous year growth; while, the total increment up to a given age divided by that age is known as mean annual increment. The timber weight and total biomass of theses poplar stands were estimated using the regression model developed by Dhanda and Verma [30] for *Populus deltoids* by taking 6-year age as rotation period, in which the destructive sampling was used.

$$\text{Volume}_{(OB)} = 0.00703 + 0.32223 * (D^2H) \qquad \text{Weight} = 268.806 \, (D^2H)^{0.976581}$$

$$(R^2 = 0.9650, \ SE = 0.02926), \qquad (R^2 = 0.9831 \text{ and } r^2 = 0.970)$$

Where, OB denotes over bark; D for diameter at breast height, i.e., 1.37 m; H for tree height.

### Statistical analysis

The statistical analysis was carried out using the completely randomized block design (CRBD) protocol for tree morphometric data, while completely randomized design (CRD) was used to estimate the physicochemical characteristics of three textured soils [31]. Data was analyzed using the SPSS statistical package, the data were statistically analyzed using a two-way ANOVA ($p < 0.05$). Two sources of variation were studied, i.e., Poplar stand planted on varying soil texture classes and age of trees. Normality of data was tested with Kolmogorov-Smirnov test and homoscedasticity with Levene's test. All possible pairs of treatments means, *i.e.,* physicochemical characteristics of soils, were compared with Duncan's multiple range test (DMRT) at 5% probability level [32]. The DMRT is similar to that of *lsd* test. DMRT involves the computation of numerical boundaries that allow the classification of the difference between any two treatment means as significant or non-significant. Spearman's correlation analysis was used to investigate the relationships between soil properties and growth parameters of poplar trees.

**Table 2. Physicochemical properties of study sites with reference to soil texture and depth of soil.**

| Soil properties | 0-30 cm | | | | 30-60 cm | | | | 60-90 cm | | | |
|---|---|---|---|---|---|---|---|---|---|---|---|---|
| | Loamy sand (PS1) | Silt loam (PS2) | Sandy loam (PS3) | *p* value | Loamy sand (PS1) | Silt loam (PS2) | Sandy loam (PS3) | *p* value | Loamy sand (PS1) | Silt loam (PS2) | Sandy loam (PS3) | *p* value |
| Sand (%) | 74.0 (3.19)[b] | 46.1 (1.89)[c] | 77.3 (2.54)[a] | 0.14* | 76.3 (1.78)[a] | 50.3 (1.46)[b] | 74.9 (2.04)[a] | 0.15* | 84.2 (1.24)[a] | 80.0 (1.64)[b] | 71.7 (1.78)[c] | 0.08* |
| Silt (%) | 13.9(1.02)[b] | 34.5 (2.15)[a] | 7.5 (1.72)[c] | 0.011* | 13.4(1.47)[b] | 46.5 (4.56)[a] | 8.1 (0.75)[c] | 0.014* | 11.8 (1.23)[b] | 6.4 (0.46)[c] | 24.3 (1.43)[a] | 0.018* |
| Clay (%) | 12.1(1.14)[b] | 19.4 (1.45)[a] | 14.8 (0.78)[b] | 0.003** | 10.4 (1.72)[b] | 3.2 (0.72)[c] | 17.0 (2.49)[a] | 0.001** | 4.0 (0.62)[b] | 13.6 (1.41)[a] | 4.0 (0.41)[b] | 0.002** |
| pH | 8.77(0.1)[a] | 8.59 (0.2)[a] | 7.87 (0.1)[a] | 0.10 | 8.89 (0.2)[a] | 8.63 (0.1)[a] | 7.65 (0.1)[a] | 0.19 | 8.84 (0.2)[a] | 8.81 (0.1)[a] | 7.62 (0.1)[a] | 0.12 |
| EC (ds m⁻¹) | 0.17 (0.25)[c] | 0.24 (0.21)[a] | 0.09 (0.09)[c] | 0.001** | 0.14 (0.11)[b] | 0.21 (0.07)[a] | 0.09 (0.05)[c] | 0.001** | 0.13 (0.11)[b] | 0.17 (0.13)[a] | 0.10 (0.07)[c] | 0.001** |
| SHC (mm/sec) | 6.65(0.19)[c] | 19.04 (0.54)[a] | 10.56 (0.21)[b] | 0.011* | 1.56 (0.10)[b] | 1.82 (0.014)[b] | 4.42 (0.24)[a] | 0.009* | 0.40 (0.02)[b] | 3.54 (0.20)[a] | 0.60 (0.03)[b] | 0.004* |
| OM (%) | 0.59 (0.032)[c] | 0.72 (0.012)[b] | 1.03 (0.014)[a] | 0.017* | 0.15 (0.021)[c] | 0.26 (0.23)[b] | 0.57 (0.028)[a] | 0.012* | 0.10 (0.011)[b] | 0.26 (0.015)[a] | 0.15 (0.016)[b] | 0.005* |
| OC (%) | 0.44 (0.02)[b] | 0.45 (0.1)[b] | 0.49 (0.02)[a] | 0.000** | 0.22 (0.01)[b] | 0.20 (0.01)[b] | 0.23 (0.02)[a] | 0.000** | 0.11 (0.01)[a] | 0.10 (0.01)[b] | 0.09 (0.01)[b] | 0.000** |
| N (kg ha⁻¹) | 108.71 (4.56)[c] | 143.69 (5.78)[b] | 196.93 (11.89)[a] | 0.032* | 75.19 (4.85)[b] | 93.30 (5.12)[b] | 168.44 (8.23)[a] | 0.027* | 91.10 (4.29)[b] | 129.57 (7.26)[a] | 87.80 (46.87)[b] | 0.014* |
| P (kg ha⁻¹) | 15.65 (3.45)[a] | 12.55 (2.47)[b] | 15.01 (2.12)[a] | 0.002** | 10.57 (3.42)[a] | 6.68 (2.01)[b] | 1.73 (0.087)[c] | 0.001** | 7.85 (1.47)[a] | 5.31 (0.92)[b] | 0.43 (0.04)[c] | 0.004** |
| K (kg ha⁻¹) | 184.76 (18.27)[b] | 187.07 (17.98)[b] | 430.66 (39.25)[a] | 0.047* | 130.19 (11.24)[b] | 145.95 (7.89)[b] | 277.91 (20.45)[a] | 0.034* | 128.38 (10.26)[b] | 198.05 (24.16)[a] | 116.38 (13.45)[b] | 0.041* |

**Note:** Standard errors are in brackets; Significant differences are indicated with *$p < 0.05$ and **$p < 0.01$ (n = 3); Different letters indicate significant differences between three stands in each depth.

**Note:** EC – Electrical conductivity; SHC – Saturated hydraulic conductivity; OM – Organic matter; OC – Organic carbon.

## Results

Because of differences in soil depth and intrinsic soil particle compositions, the physicochemical characteristics of three sites having different soil textural classes selected for establishment of poplar stands varied significantly (Table 2). Compared to PS1 and PS3, PS2 has substantially more silt, clay, EC, and SHC. PS1, on the other hand, has significantly more P only, while, PS3 had sand fraction at surface soil, *i.e.,* 0–30 cm of depth; contents of organic matter, and macronutrients (NK). PS1 and PS2 had sandy loam and silt loam soil textures, respectively, whereas PS3 had loamy sand soil texture. In all three depth ranges, these variations were noted (0–30, 30–60, and 60–90 cm). In the sub-surface soil or depth of 30–60 cm, PS3 had significantly higher clay content and SHC, while, PS1 had P mineral only throughout the soil profile.

The present study examining the growth behaviour and productivity potential of poplars across seven age-gradations revealed statistically significant variations among the poplar stands. Up to 5-years of growth, the tree height and diameter at breast height (dbh) were significantly higher in PS3 stand and after that in PS2 (Fig 1) compared to PS1 ($p < 0.01$). Average leaf area and crown spread were significantly higher in PS2 from the establishment age to 7th year age. Significantly similar trend was observed for the timber weight and total biomass as in diameter growth. The growth and

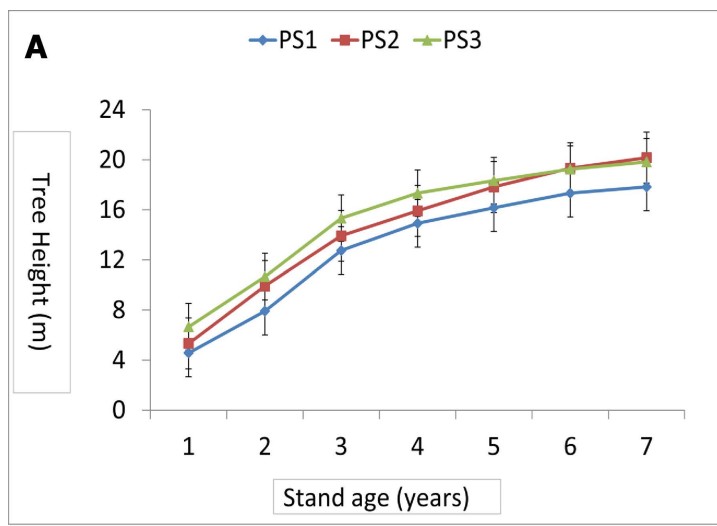

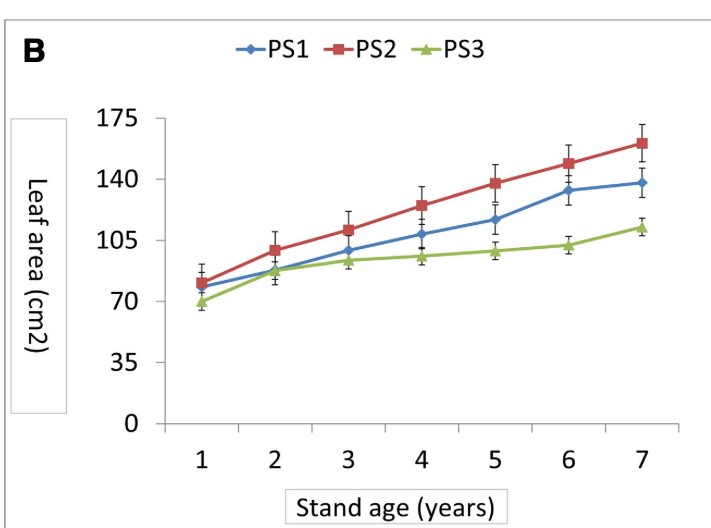

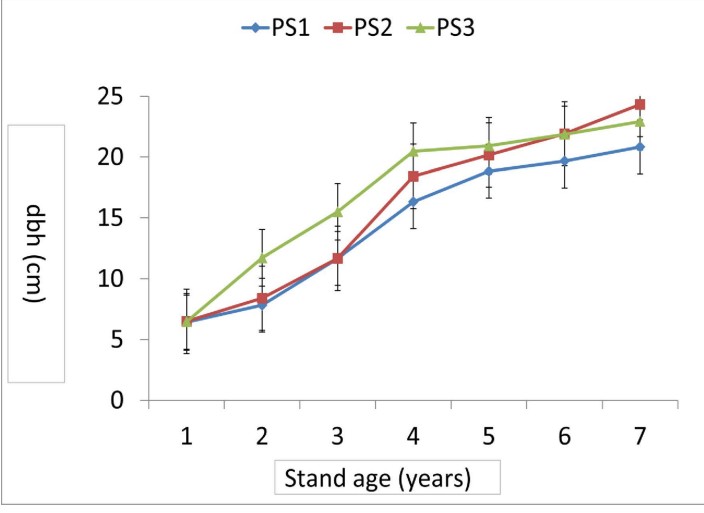

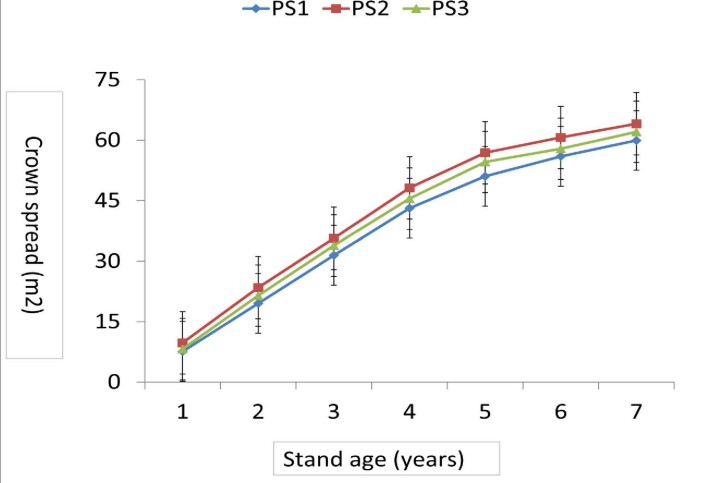

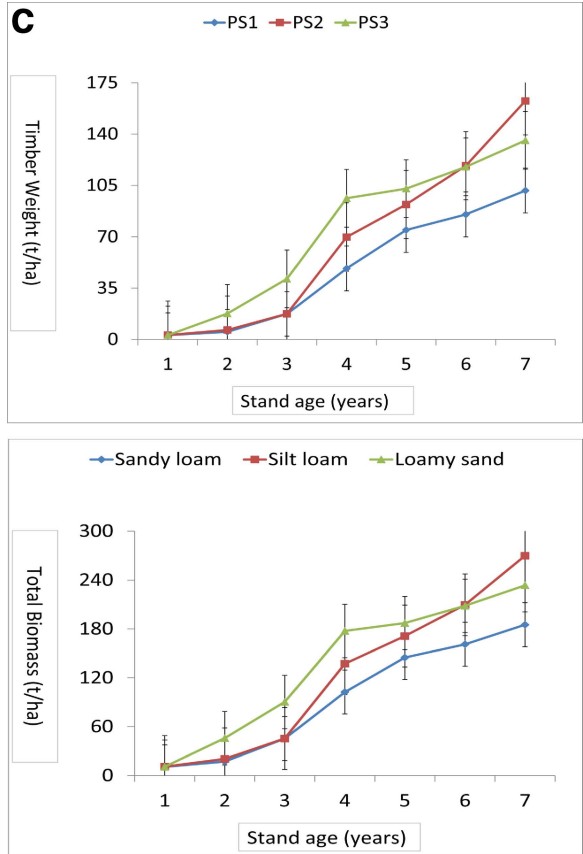

**Fig 1. Growth parameters and productivity of three poplar stands over seven age-gradations; Error bars are standard errors.**

productivity of poplar stands increased with the stand's age (Fig 1). CAI for tree height, dbh, timber weight and total biomass was increased up to 3$^{rd}$ age gradation followed by decreased (Fig 2). However, CAI for timber weight and total biomass in PS2 grown in silt loamy texture soil was increased. Comparison of mean annual increment (MAI) of stand height, dbh, timber weight and total biomass at seven age-gradations showed that higher values were in PS3 up to 5$^{th}$ age-gradation after that in PS2 (Fig 2). The MAI for tree height and dbh were higher in initial age-gradations, while, timber weight and total biomass were significantly higher in mid-rotation age-gradations.

The results of Pearson correlation coefficient among growth parameters and soil characteristics of poplars were illustrated in Table 3. Fig 3 illustrated scatter plots with regression lines to show the evidence the relationships among Poplar growth parameters and soil properties. The amount of clay percent, SHC, organic matter, OC, soil N, P and K, and leaf area had significant positive correlations with tree height, diameter and timber weight except the silt percent, EC. Organic carbon and soil P had very strong correlation with timber weight, while, organic matter and soil K showed low correlations in poplar stands.

## Discussion

### Physicochemical properties of soils

Any soil can be kept in good condition and provide nutrients needed for the development and growth of any type of tree by altering the ratios of sand, silt, and clay within a soil. The primary factor that affect the soil's ability to hold water include

its texture, which also affects other variables that eventually affect root growth, such as pH, buffering capacity, organic matter content, drainage, aeration, saturated hydraulic conductivity, EC, and cation exchange capacity [33]. Significant differences were noted in the physicochemical characteristics due to the changes in soil particulate composition in soil profile (Table 2). The effect of soil texture on soil nutrients is caused by changes in soil environmental factors, such as pH, soil hydraulic properties, EC, water-holding capacity, which affect soil microbial system composition, activity, and function, thereby impacting nutrient cycling and their availability [34].

As the depth of the soil increased in the current study, the proportion of sand, silt, and clay also changed, changing the textural class. For example, at 60–90 cm, sandy loam changed to loamy sand, at 30 cm, silt loam changed to sandy loam, and at 15 cm, loamy sand changed to sandy loam once more in to loamy sand at 60 cm depth. In the present study, the distributions of soil particle sizes varied significantly with soil depths in all soil textural classes (Table 2). Poplar plantations experienced variable annual growth increments with age due to these changes in soil particle proportions as result different soil physicochemical parameters changed and affecting the tree growth [18]. According to reports by Dhaliwal et al. [35] in Indian Punjab, Asmamaw and Mohammed [36] in Ethiopia, these variances may be the result of variations in the land-covers covering the surfaces. These variations in soil texture could affect the availability of nutrients at different

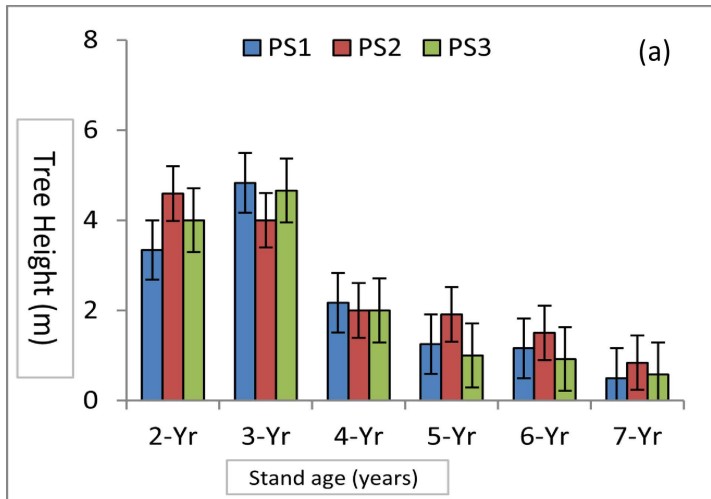

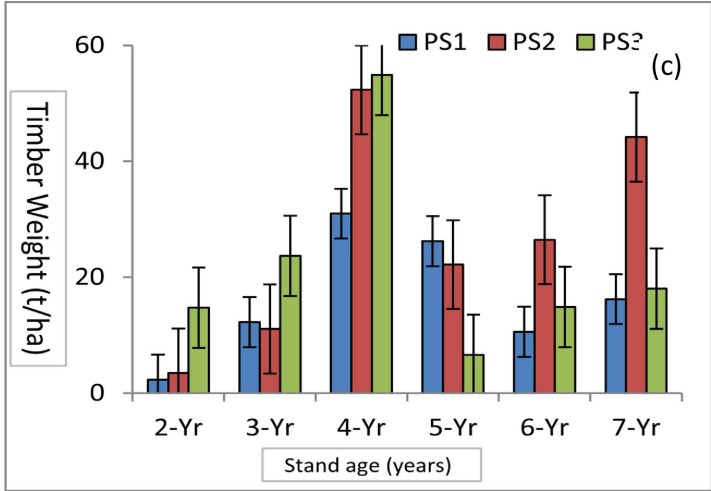

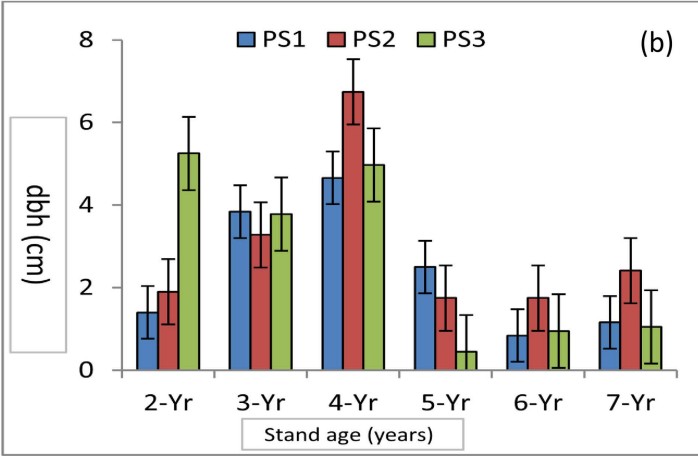

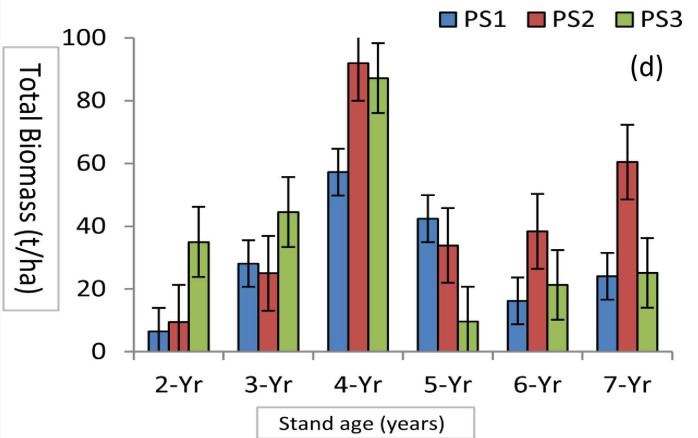

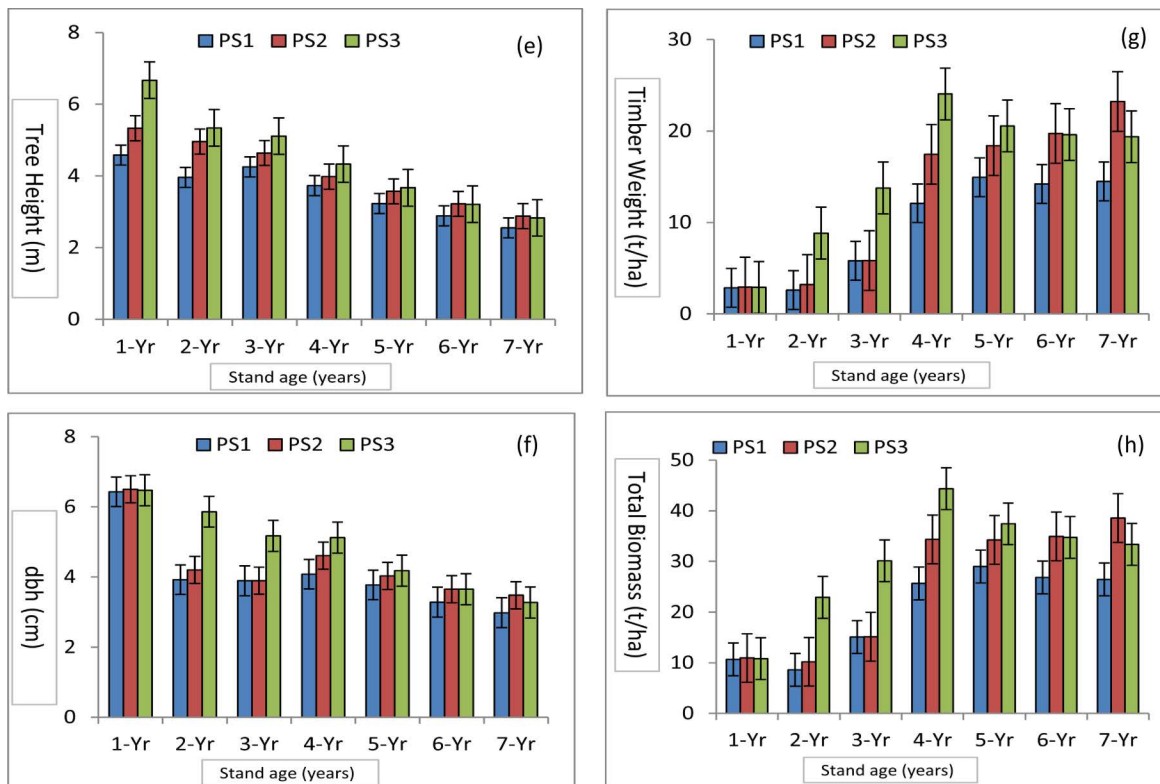

**Fig 2. Annual increments: CAI (a-d) and MAI (e-h) for tree height, dbh, timber weight and total biomass of three poplar stands for seven age-gradations; Error bars represent standard errors.**

**Table 3. Correlations between soil properties and growth parameters of poplar stands.**

| Soil properties | $H_{tree}$ | *p* value | dbh | *p* value | $W_{tim}$ | *p* value |
|---|---|---|---|---|---|---|
| Silt (%) | −0.675 | 0.004** | −0.513 | 0.027* | −0.543 | 0.019* |
| Clay (%) | 0.784 | 0.000** | 0.649 | 0.006** | 0.672 | 0.004** |
| EC (ds m⁻¹) | −0.791 | 0.001** | −0.545 | 0.024* | −0.597 | 0.011* |
| SHC (mm/sec) | 0.425 | 0.024* | 0.612 | 0.008* | 0.678 | 0.007* |
| Organic matter (%) | 0.523 | 0.021* | 0.429 | 0.017* | 0.473 | 0.012* |
| Organic carbon (%) | 0.800 | 0.004* | 0.835 | 0.002** | 0.842 | 0.003** |
| Soil N (kg ha⁻¹) | 0.583 | 0.003** | 0.572 | 0.009** | 0.601 | 0.007** |
| Soil P (kg ha⁻¹) | 0.782 | 0.004** | 0.813 | 0.002** | 0.842 | 0.002** |
| Soil K (kg ha⁻¹) | 0.485 | 0.016* | 0.460 | 0.016* | 0.421 | 0.023* |
| Average leaf area | 0.763 | 0.007** | 0.763 | 0.003** | 0.714 | 0.004** |

**Note:** $H_{tree}$: tree height; dbh: diameter at breast height; $W_{tim}$: timber weight; Significant differences are indicated with *$p < 0.05$ and **$p < 0.01$ (n = 3).

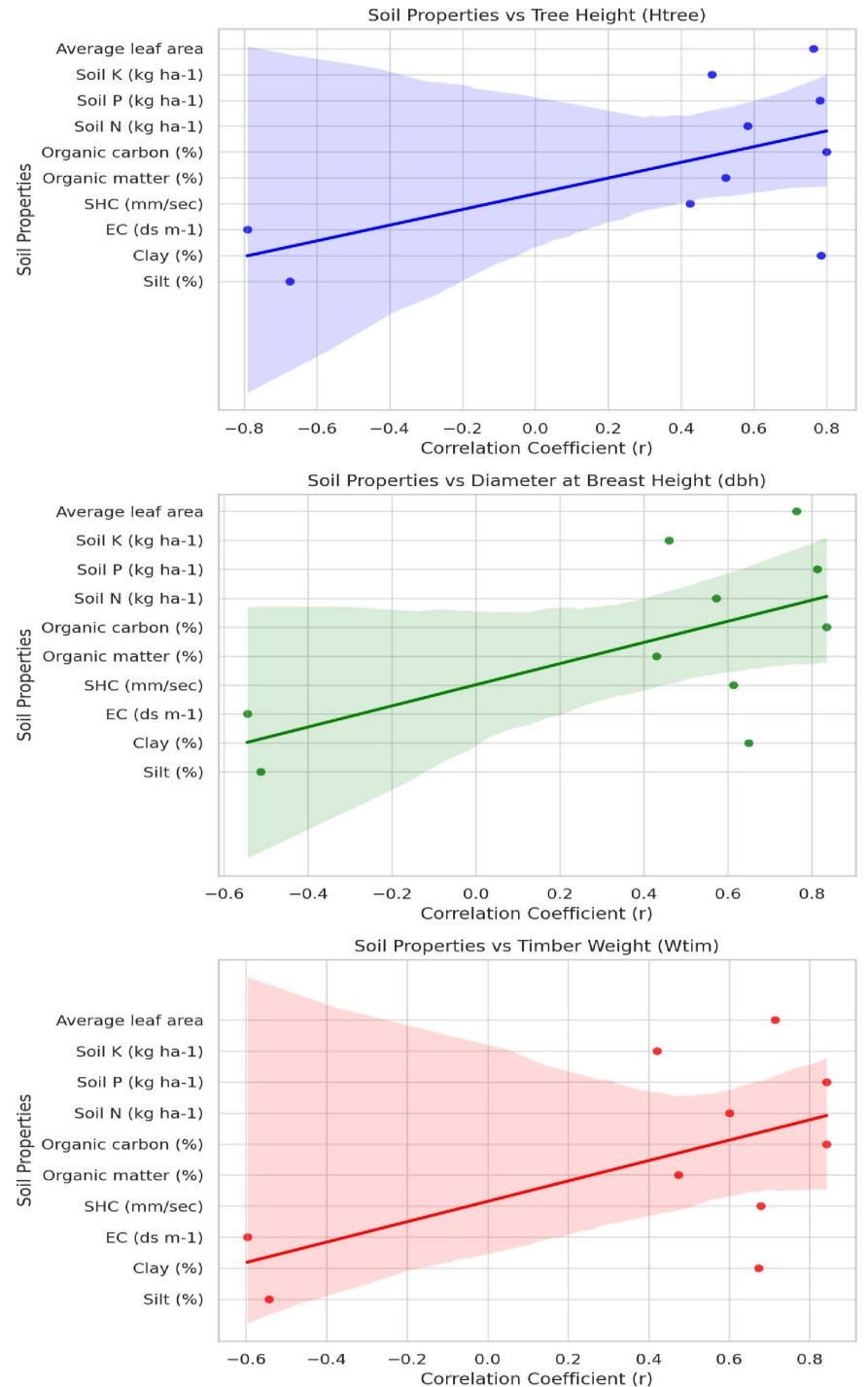

**Fig 3. Visual illustrations of Spearman's correlation analysis to investigate the relationships between soil properties and growth parameters.**

depths, which could modify how roots grow. According to Pinno et al. [37], a sand concentration of 55–70% was shown to be optimal for poplar development. Pinno and Belanger [38] also discovered that the best indicator of poplar growth was soil texture, with more silt and clay producing better growth.

Changes in the proportionate distribution of soil particles in corresponding textural class may be the cause of changes in soil pH, EC, and SHC with depth. An increasing trend was observed with an increase in soil depth for sandy loam and silt loam soil, while decreasing trend was observed in loamy sand soil (Table 2). Because topsoil is rich in organic matter and decomposition of that matter produces more organic acids, which lower pH, the pH value of topsoil is generally lower in all texture classes [39]. Nonetheless, statistical analysis revealed that there was minimal variation in soil pH at different depths under age-graded poplar plants. The presence of iron and manganese ions may account for pH values in sandy loam soil profiles' tendency to rise with depth [40]. For all soil depths, the silt loam, sandy loam, and loamy sand soils had the lower EC values which is the measure of salinity level. This declining trend was only seen in silt loam and loamy sand soils as soil depth increased. Similar to Faruque et al. [41], higher EC in silt loam soils was caused by an increase in clay content. In all texture classes, SHC exhibited a declining tendency as soil depth increased, which may have resulted from modifications to water conductivity and capillary water suction patterns [42].

## Poplar productivity

Numerous studies have assessed the productivity and growth performance of different poplar species in varying climatic conditions and management practices. On the other hand, effect of soil particle compositions altering the growth behaviour of poplar plantations with age is still not well recorded. Three distinct poplar plantations differ significantly in terms of height, dbh, timber weight and total biomass during the course of seven-year rotation in poplar trees. The mean annual increment (MAI) for *P. deltoides* was 34.39, 28.23, and 22.67 $m^3ha^{-1}yr^{-1}$ in PS3, PS2 and PS1, respectively which was higher than reported from 13−33 $m^3ha^{-1}yr^{-1}$ over 5-year rotation in *P. Nigra* in Iran [43] and 25.4 $m^3ha^{-1}yr^{-1}$ at 7 years of age for *P. tremula × P. tremuloides* in northern Poland [44]. The current study's findings demonstrated that poplar trees in PS3 had above-ground productivity of 42.40 $m^3ha^{-1}yr^{-1}$ that was considerably higher than that of PS2 (38.80 $m^3ha^{-1}yr^{-1}$) having loamy sand texture and PS1 (32.83 $m^3ha^{-1}yr^{-1}$) having silt loam texture when we considered 5 age-gradations as rotational age. However, poplar stand showed higher productivity in PS2 (43.70 $m^3ha^{-1}yr^{-1}$) than PS3 (37.80 $m^3ha^{-1}yr^{-1}$) and PS1 (29.97 $m^3ha^{-1}yr^{-1}$). It was due the change in the soil particle composition and an increase in the silt content in PS3 site below the 60 cm depth. After 60 cm depth, the soil profile in PS2 is almost similar with site PS3 in soil particle composition, however, the texture class was same as per the soil classification. The present study suggests that the higher biomass productivity in sandy loam may be attributed to the plantation's lower sand content (approximately 50%) in the root zone.

The growth performance and productivity of poplar trees are influenced by various factors, including the quality of the one-year-old seedlings, planting geometry, factors of locality including, fertilization and irrigation regimes, and management approaches [9,10,45] and these three poplar stands had also the same age clonal planting material, package and practices of cultivation. Instead, poplar stands were practically adjacent (a distance of 2.85 and 7.30 km), meaning that the environmental parameters, such as topography and climate, were similar. The growth performance of poplar trees appears to be influenced by the qualities of the soil. Despite the three stands being next to one another, analysis of the soil attributes revealed that there were substantial differences in soil texture, organic matter, EC, and macronutrient (N, P, and K) concentrations among the poplar stands (Table 2). Heterogeneity of soil properties can occur on a big or small scale, as Feng et al. [46], and Weindorf and Zhu [47] have described.

The physical characteristics of soil affect plant growth parameters because of root penetration, aeration, and water retention properties [48]. Of which, one of the most crucial aspects of the physical characteristics of the soil for poplar plants is soil texture [49]. Based on the results, the soil texture of these poplar stands was totally different with respect of soil-depth and locality, however they are adjacent. The soils of poplar stands (PS1, PS2 and PS3) fall under the category of Sandy loam, Silt loam and Loamy sand, respectively. Our findings demonstrated that poplar trees grew more

successfully in sandy loam soil than in silt or loamy sand. It has been noted that poplars need medium-textured, deeply drained soils [50]. But scientific research on this species in flood plain areas of Indian Punjab shows that besides the aforesaid finding, to obtain its best yield up to rotation of 6 or 7 years, the bulk density of soil should be 1.5 gm cm$^{-3}$ [51]. Further studies on nutrient requirement showed that there was a huge demand of NPK and micronutrients from the 2$^{nd}$ year onward with the expansion of crown and the 3$^{rd}$ year was crucial because of its maximum dbh growth [52].

Keeping in view the growth rhythms of *Populus deltoides*, NPK fertilizer's doses on sandy loam soils in Punjab were worked out [20]. However, in the present study, this has also been seen that various texture of the soil have significantly different levels of NPK and other physicochemical parameters. On the other hand, farmers have developed the mindset related to its low nutrient requirement. This hypothesis has been rejected by the finding of the present study. It was observed that the planting of poplar entire transplants in 1m$^3$ pit had better growth above as well as below ground in comparison to traditional auger hole planting with 1m depth and 15cm diameter [53]. Therefore, the present research further describes the need of pit management at the time of planting taking into consideration the root spread information available, *i.e.,* minimum 1m radius and 1m deep to be filled with approximately 75−80% sand, 6−8% silt, and 13−17% clay soil. Packing of the soil in the pit with bulk density of 1.5 gmcm$^{-3}$ and addition of 50g urea, 85g DAP (or 85g urea and 250g SSP) to promote root development and apply 25kg zinc sulphate monohydrate (33% Zn) per acre in zinc deficient soils at the time of planting [20].

Based on the study carried out under greenhouse conditions, Daneshvar and Modirrahmati [54] classified poplars into four groups based on how well they could tolerate soil salinity: *P. deltoides* (highly sensitive), *P. euramericana* (sensitive), *P. nigra* (semi-sensitive), and *P. alba* (slightly sensitive). Poplar species vary in their ability to withstand salt; later on *P. euphratica* was reported to be the most tolerant by Silva et al. [55]; Chen and Polle [56] studies. According to Mao et al. [57], diameter and height increments decreased as soil salinity increased for *P. Alba* and *P. nigra*. As expected, the present results confirmed that *Populus deltoides* growth parameters were found significantly negatively correlated with soil EC or salinity level and showed that poplar tree productivity and growth performance was decreased in a soil having EC of 0.21 dS/m in PS2 as opposed to a soil with EC of 0.09 dS/m in PS3. The current data also revealed that *P. deltoides* is particularly susceptible to EC or salinity level during its early stages of growth (up to 4 or 5 years), and that its sensitivity to EC decreases as the root system becomes more established and penetrates into deeper layers. On the other side, the factor restricting growth in agroforests and short-rotation tree plantations is soil nutrients which is indirectly affected by soil textural class [58]. The findings of present study showed that poplars achieved more growth and productivity in the soil that contains more macronutrients (N, P, and K) and organic carbon. There was also a statistically significant positive correlation between growth parameters and nutrients. Additionally, poplars have been observed to reach their maximum growth potential on soils that are both rich (high nutrients) and well-drained (sandy loam soil) [50,59].

The present study, however, has some limitations, including: (i) Site-specific constraints, as the study was limited to a specific microclimate, making the yield patterns not universally applicable to regions with different climatic variables and edaphic conditions; (ii) Genetic variability, where differences in genetic traits among other *Populus* species may influence growth responses, making it difficult to attribute variations solely to soil properties; (iii) Variations in silvicultural practices, such as irrigation, fertilization, and spacing, which may introduce uncontrolled factors affecting growth and yield outcomes across sites; and (iv) Unaccounted biotic factors, including the potential influence of pests, diseases, and interspecies competition, which were not the primary focus of this study but may contribute to variations in growth dynamics. These limitations highlight areas for future research and help contextualize the study's findings within its defined scope.

## Conclusions

Over a period of seven age-gradations, the study showed that poplar trees (*P. deldoides*) grown on sandy loam soil texture, which has a content of approximately 75–80% sand, 6–8% silt, and 13–17% clay particles with optimal levels of macronutrients, organic matter, low electrical conductivity, and a balanced particle composition, is crucial to achieve the maximum poplar productivity. Results indicated that soil parameters significantly influenced poplar growth in both surface

and deeper soil layers. Regular soil testing and adjustments to cultivation practices based on soil conditions are recommended to improve long-term yields. Furthermore, poplar plantations on loamy sand soil were shown to produce higher total biomass and mean annual increments, making them suitable for the pulp and paper industry. Farmers typically follow 4–5 year rotation for pulp production, yielding early returns, and 6–7 year rotation for timber, which offers higher per-quintal prices due to increased diameter, suitable for timber and packaging material. For optimal monetary returns, farmers with sandy loam soils are advised to apply micronutrients (N, P and K) according to Punjab Agricultural University's recommendations to achieve a target plant girth of 90 cm.

## Author contributions

**Conceptualization:** Ashok Kumar Dhakad, Avtar Singh.

**Data curation:** Ashok Kumar Dhakad, Tarsem Singh, Avtar Singh, Kuldip Singh.

**Formal analysis:** Tarsem Singh, Kuldip Singh.

**Investigation:** Ashok Kumar Dhakad, Avtar Singh.

**Methodology:** Ashok Kumar Dhakad, Tarsem Singh, Avtar Singh, Kuldip Singh.

**Project administration:** Ashok Kumar Dhakad.

**Resources:** Ashok Kumar Dhakad, Tarsem Singh, Avtar Singh, Kuldip Singh.

**Software:** Ashok Kumar Dhakad, Tarsem Singh.

**Supervision:** Ashok Kumar Dhakad, Avtar Singh.

**Validation:** Ashok Kumar Dhakad.

**Visualization:** Ashok Kumar Dhakad.

**Writing – original draft:** Ashok Kumar Dhakad.

**Writing – review & editing:** Ashok Kumar Dhakad, Tarsem Singh, Avtar Singh.

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
