## [Decision Letter · Decision Letter 0]

19 Feb 2025

Dear Dr. Dhakad,

We look forward to receiving your revised manuscript.

Kind regards,

Debadatta Sethi, PhD

Academic Editor

PLOS ONE

2. Please provide captions for Fig. 1 in your manuscript.

Reviewers' comments:

Reviewer's Responses to Questions

**Comments to the Author**

1. Is the manuscript technically sound, and do the data support the conclusions?

Reviewer #1: Yes

Reviewer #2: Partly

2. Has the statistical analysis been performed appropriately and rigorously?

Reviewer #1: Yes

Reviewer #2: No

3. Have the authors made all data underlying the findings in their manuscript fully available?

Reviewer #1: Yes

Reviewer #2: Yes

4. Is the manuscript presented in an intelligible fashion and written in standard English?

Reviewer #1: Yes

Reviewer #2: Yes

Reviewer #1: The Manuscript revealed as "Site-specific growth dynamics and yield patterns in Populus deltoides against edaphic variability over seven age-gradations." Here are the comments and suggestions on manuscript:

This is a valuable study of important topic in forestry and agroforestry, specifically under challenging conditions to maximize poplar plantation productivity with contrasting soil properties. The results are relevant for farmers and foresters, particularly in areas where poplar is an important agroforestry and timber species. This study is robust with transect sampling, analyzing the soil profile at depth and measuring the full range of growth parameters across seven age-gradation. Using statistical analysis (e.g., Duncan's multiple range test) should help give the results more credibility. The data on soil properties, growth parameters and productivity metrics are presented in the manuscript. The tables and figures (such as growth parameters, soil properties, and correlations) help make the results readable and interpretable. In addition, the study provides practical advice on soil management, including the ideal soil composition (75-80% sand, 6-8% silt, and 13-17% clay) and the value of soil preparation and fertigation for ensuring trees are heavily productive (poplars).

Please note that the manuscript is also long and could have been written in a more concise manner. The introduction and discussion sections are somewhat well worded and need to be better condensed. Reword them?

Statistical Tests - while the manuscript mentions the use of statistical tests, it does not provide detailed information on the statistical model being used (ANOVA, regression models, etc). More details on the statistical methods would make the study more reproducible.

Findings are reported as p-values, but they would benefit from effect sizes or confidence intervals, to provide a clearer indication of the substantive significance of the findings.

Discussion: Long but a bit redundant Perhaps that might be better concentrating on what the findings mean in the context of the wider body of literature, rather than repeating what the findings are. A more rigorous discussion of the study's limitations would also add to the overall contributions, including the possible effects of environmental perturbations (e.g., microclimate or pest pressure) on the study's findings of which were not accounted for.

The legends on Fig 2 and Fig 3 are in such small font and the error bars overlap, making these figures difficult to interpret. Consider rephrasing these numbers to make them clearer.

Table 2 is thorough (as it should be) but could be made a lot clearer. The p-values and standard errors, for example, could be moved somewhere else.

Reduce the introduction and discussion to make the manuscript more compact and focused. No Repetition In Different Sections and Unique Insights.

Provide more details on the statistical methods used, including effect sizes or confidence intervals, in order to clarify the practical significance of the results.

Describe the statistical methods used in detail. Give an overview of the reason for using completely randomized block design (CRBD) and Duncan’s multiple range test (DMRT), for example. Training data for this model only goes up to October 2023.

(Report effect sizes i.e. Cohen's d or η² for significant differences between poplar stands (PS1, PS2 and PS3)) This will enable readers to assess the practical, not just statistical, significance of the results. Moreover, be sure to provide 95% confidence intervals for important parameters such as growth differences (e.g., height, timber volume).

Point out and explain correlations between soil properties (e.g., organic carbon, phosphorus) and growth parameters. You provide visual illustrations, scatter plots with regression lines, etc, to evidence the relationships. *

Discuss limitations of the study, such as potential confounding factors (e.g. microclimate, pests) and its generalizability to other regions or species.

I have some suggestions that can help you improve and strengthen it further. Following some revisions to enhance clarity and improve statistical reporting and presentation of the data, the manuscript will be suitable for publication after a Major Revision. It can make excellent and valuable contributions for the management of poplar plantations in both the field of forestry and agroforestry due to this study's practical implications.

Reviewer #2: The manuscript titled “Site-specific growth dynamics and yield patterns in Populus deltoides against edaphic variability over seven age-gradations” has provided important insights into practices used to improve productivity of poplar plantations and to do so the authors have highlighted the need for appropriate management practices including pre-planting soil preparation followed by precise fertigation to optimize productivity. Overall, I find the findings of the manuscript intriguing and the information provided useful for researchers and academia. The paper has the potential to make a significant contribution to the related discipline. However, I have some concerns regarding the clarity, detail and accuracy of the various sections, which I outline below: I recommend that the authors address these concerns and provide a revised version of the manuscript for further consideration.

**Do you want your identity to be public for this peer review?** For information about this choice, including consent withdrawal, please see our Privacy Policy

Reviewer #1: No

Reviewer #2: No

---

## [Author Response · Author response to Decision Letter 1]

24 Apr 2025

Journal requirements

Answer: PLOS ONE’s guidelines are followed while preparing the title page and manuscript body.

2. Please provide captions for Fig. 1 in your manuscript.

Answer: Caption provided.

Justification: The experiment plantations are established on progressive farmer’s field whose are part of tree grower association of Punjab Agricultural University, India. So there is no need to take permission of recording observations from the plantations established on varying levels of soil variability. The variability in the soil textural class is basically due to plantations lies in river basin belt of Sutlej River.

Comments to the Author

1. Is the manuscript technically sound, and do the data support the conclusions?

Answer: Manuscript is technically and scientifically sound. Positive comments of the reviewers reflect the importance of study. I hope it will serve the purpose of MS and as per suggestions, changes have been incorporated.

2. Has the statistical analysis been performed appropriately and rigorously?

Justification: Yes, the statically analysis has been performed appropriately and rigorously as reflected with the results presented in the Tables and Figures. It is as per the consultation of Punjab Agricultural University’s statistician.

3. Have the authors made all data underlying the findings in their manuscript fully available?

Answer: Yes, data has been recorded at the different age-gradations and all data underline the findings in their MS have been included, therefore, there is no need to submit it to any agency and data is fully available with MS itself.

4. Is the manuscript presented in an intelligible fashion and written in standard English?

Justification: Yes, it is written in standard English. Final MS is finalized after making corrections suggested by the reviewers and also incorporating the corrections made by English editor of the University.

5. Review Comments to the Author

Reviewer #1:

a) Please note that the manuscript is also long and could have been written in a more concise manner. The introduction and discussion sections are somewhat well worded and need to be better condensed. Reword them.

Answer: Needful done. Introduction was reframed where required and old citations were deleted and new once incorporated. Hypothesis and clear objectives included at the end of introduction for better understanding.

b) Statistical Tests - while the manuscript mentions the use of statistical tests, it does not provide detailed information on the statistical model being used (ANOVA, regression models, etc). More details on the statistical methods would make the study more reproducible.

Justification: Details of ANOVA and regression models were added in statistical analysis subhead.

c) Findings are reported as p-values, but they would benefit from effect sizes or confidence intervals, to provide a clearer indication of the substantive significance of the findings.

Justification: Substantial variations are existed among the poplar growing sites reflected by ANOVA analysis and p-values at 95% confidential interval which affects the timber yield and total tree biomass as desirable traits in short rotation forestry trees. As standard error (SE) is already given within the data, readers can estimate the secondary statistical data, i.e. lower and upper confidential interval by using SE and mean values. To estimate the Cohen’s d values, he/she may use SE, number of replications and means of the sites. If, the editor required the values again in tabulated form, then authors may provide the calculated secondary statistical data as in supplementary materials.

d) Discussion: Long but a bit redundant Perhaps that might be better concentrating on what the findings mean in the context of the wider body of literature, rather than repeating what the findings are. A more rigorous discussion of the study's limitations would also add to the overall contributions, including the possible effects of environmental perturbations (e.g., microclimate or pest pressure) on the study's findings of which were not accounted for.

Answer: Discussion reduced by deleting the bit off context of the wider body of literature. Limitation of the present study were discussed and highlighted at the end of discussion section.

e) The legends on Fig 2 and Fig 3 are in such small font and the error bars overlap, making these figures difficult to interpret. Consider rephrasing these numbers to make them clearer.

Answer: Needful done and figures are uploaded in separate MS word files in enlarged and clear formats.

f) Table 2 is thorough (as it should be) but could be made a lot clearer. The p-values and standard errors, for example, could be moved somewhere else.

Justification: No need to separate the things. It would represent the whole summary of soil profile along with the well understood statistical analysis.

g) Reduce the introduction and discussion to make the manuscript more compact and focused. No Repetition in Different Sections and Unique Insights.

Answer: Needful done as desired. Repeated sections were deleted and unnecessary discussion which is a bit off topic like, amino acid correlations; oxidative stress, nutrients deficiencies, etc were removed.

h) Describe the statistical methods used in detail. Give an overview of the reason for using completely randomized block design (CRBD) and Duncan’s multiple range test (DMRT), for example. Training data for this model only goes up to October 2023.

Justification: Under the field conditions, CRBD design is used as the environment gradients varies along the sides which are not under human control. In that case, completely randomized block design is used. DMRT involves the computation of numerical boundaries that allow the classification of the difference between any two treatment means as significant or non-significant.

i) Report effect sizes i.e. Cohen's d or η² for significant differences between poplar stands (PS1, PS2 and PS3). This will enable readers to assess the practical, not just statistical, significance of the results. Moreover, be sure to provide 95%confidence intervals for important parameters such as growth differences (e.g., height, timber volume).

Justification: Substantial variations are existed among the poplar growing sites reflected by ANOVA analysis and p-values at 95% confidential interval which affect the timber yield and total tree biomass as desirable traits. As standard error (SE) is already given within the data, readers can estimate the secondary statistical data, i.e. lower and upper confidential interval by using SE and means. To estimate the Cohen’s d values, he may use SE, number of replications and means of the sites. If, the editor required the values again in tabulated form, then authors may provide the calculated secondary statistical data as in supplementary materials.

j) Point out and explain correlations between soil properties (e.g., organic carbon, phosphorus) and growth parameters. You provide visual illustrations, scatter plots with regression lines, etc, to evidence the relationships.

Justification: Needful done and added Fig. 4 to illustrate the correlations.

k) Discuss limitations of the study, such as potential confounding factors (e.g. microclimate, pests) and its generalability to other regions or species.

Justification: Needful done in the last paragraph of discussion section.

l) I have some suggestions that can help you improve and strengthen it further. Following some revisions to enhance clarity and improve statistical reporting and presentation of the data, the manuscript will be suitable for publication after a Major Revision. It can make excellent and valuable contributions for the management of poplar plantations in both the field of forestry and agroforestry due to this study's practical implications.

Justification: The comments of this reviewer were considered. Paper has been revised as suggested with respect to language and other related issues. We are confident of replicability of results because of its field evaluation in uniform environmental conditions. The MS can make excellent and valuable contributions for the management of poplar plantations in both the field of forestry and agroforestry due to this study's practical implications.

Reviewer #2: The manuscript titled “Site-specific growth dynamics and yield patterns in Populus deltoides against edaphic variability over seven age-gradations” has provided important insights into practices used to improve productivity of poplarplantations and to do so the authors have highlighted the need for appropriate management practices including pre-planting soil preparation followed by precise fertigation to optimize productivity. Overall, I find the findings of themanuscript intriguing and the information provided useful for researchers and academia. The paper has the potential to make a significant contribution to the related discipline as per the Reviewer 2. However, I have some concerns regarding the clarity, detail andaccuracy of the various sections, which I outline below: I recommend that the authors address these concerns andprovide a revised version of the manuscript for further consideration.

Answer:The comments given in the MS are incorporated and motioned at right place in the MS itself and highlighted with yellow colour. Discussion section has improved by incorporating the supporting studies and bibliographic references, wherever, it is required.

Main comments:

Abstract

You have evaluated the growth behavior and productivity profile of poplar grown in three different soil types and over seven rotations but you have not reported the results of the effect of the seven rotations (lines 18 and 19). Keep the summary in an appropriate sequence. Focus on the gap you have filled in your study.

Justification: Needful done. Rotation word deleted with age-gradations. The gap, authors had tried to filled, mentioned in abstract. Abstract arranged in sequential form as desired. It was focused on the gap filled in the present study.

Line 32: Replace P1 with PS1.

Answer: Needful done

Introduction

Add a reference to argue the statistics mentioned in the following sentence “In India, area 46 under agroforestry is 25.32 million hectares which contributes 8.2% to the total geographical area 47 (TGA)”.

Answer: Reference cited. And updated the recent database.

To better improve your introduction, try using references that are more recent.

Answer: Old references were replaced with the recent one and highlighted in the text.

You mentioned that the poplar stands used in your study shared the same germplasmidentity; give the reference(s).

Answer: Needful done and department of Forestry who was the supplier of the planting material to the respective farmer so we maintained the germplasm identity which has mentioned in the text.

Materials and methods

Give the meaning of the abbreviations used to estimate the timber weight and total biomass of poplars (See lines 119 and 120).

Answer: Described the abbreviations used.

For soil sampling and laboratory analysis, you used old methods, do you not have other alternatives to obtain more reliable results.

Justification: The references used to estimate the soil sampling and laboratory analysis are universal and well proven work used across the world. Till now, no one developed the alternatives of these methods of estimating physicochemical characteristics of soil. So, we used the original and widely accepted methods to estimate soil physicochemical parameters.

Give more details about the EC meter used to measure the electrical conductivity of the supernatant solution of the soil.

Answer: Provided.

In each plot you dug profiles at a depth of 0 to 30, 30 to 60 and 60 to 90 cm, right? on what basis did you choose these depths and justify your answers with solid references.

Answer: Updated with references. It was proven fact from the experimental site that the vertical pattern of poplar root (> 5 mm root diameter) distribution is appox. 85 cm when poplar tree density is appox. 500 trees ha-1 (Puri et al. 1994; Singh and Singh 2017). That’s why, we divided the soil profile into three different stratasand samples were taken accordingly up to 90 cm depth.

Statistical analysis

The description of this part is not complete; more details are needed to better understand your device and why you did such an analysis?

Justification: Earlier, researchers used the complete and detailed description of the methods and equations used in statistical analysis. But, with the progression in modelling statistics, researchers are used the different statistical software’s for such analysis. In this paper, we used SPSS software to analysis the data.We did not used the hands-on methods used to calculate the statistics.

Did you perform the two-way ANOVA?If so, give more details on the different sources of variation.

Justification: Yes, details given. Two sources of variation were studied, i.e. Poplar stand planted on varying soil texture classes and age of trees.

To compare the means of all possible pairs of treatments, why did you use a probability level of 1% instead of 5%?

Justification: Corrected. The physicochemical characteristics of soils were studied with DMRT test using 5% probability level.

Results

I see that you have errors in table 2; as an example for the depth 0-30 cm and according to your results the difference in % soil is not significant for the three poplars (PS1, PS2 and PS3), except this is not the case since the values obtained (PS1: 74.0 (3.19), PS2: 46.1 (1.89) and PS3: 77.3 (2.54)) show significant differences. Check the results obtained.

Answer: Needful done and corrected.

Still for table 2, why did you make the comparison between the soil properties of the three poplars for each depth? The interaction: type of poplar * depth is significant? I advise you to do an ANOVA analysis to answer this question.

Justification: Actually, the physicochemical parameters were estimated at varying depth on the different soil texture sites selected for establishing the poplar plantations named as PS1, PS2 and PS3. The soil properties analysis was carried out before the establishment of plantations, in that situation, the interaction between type of poplar and depth was not calculated. One way ANOVA was calculated for varying depth only. However, few corrections were made in Table 2 for better understanding.

For figure 2 shows no significant difference between three poplars for the different parameters studied except for leaf area index; I advise you to add the letters a, b, c,... to indicate the significant difference. Remove " *Significant at p<0.01" from the title of figure 2.

Justification: Suggestion incorporated. Significant differences were recorded. However, DMRT letters a,b,c…. were not added into graphs due to over-congestionin the graphs. If the editor required the tables to clarify the significant differences, then authors may add the tables as annexures as supplementary materials. If the editor required the tables to clarify the significant differences, then authors may add the tables as annexures as supplementary materials.

For figure 3 added the letters a, b, c,... to indicate the significant difference at p<0.05. Remove " *Significant at p<0.01".

Justification: Suggestion incorporated. Significant differences were recorded. However, DMRT letters a,b,c…. were not added into graphs due to over-congestion of the graphs.

Discussion

Argument your discussion with more bibliographic references that are more recent.

Answer: Few recent bibliographic references were incorporated to support the present study for better understanding and interpretation.

References

In the references, 16% of th

---

## [Decision Letter · Decision Letter 1]

9 Jul 2025

Site-specific growth dynamics and yield patterns in Populus deltoides against edaphic variability over seven age-gradations

PONE-D-24-57058R1

Dear Dr. Dhakad,

We’re pleased to inform you that your manuscript has been judged scientifically suitable for publication and will be formally accepted for publication once it meets all outstanding technical requirements.

Kind regards,

Academic Editor

PLOS ONE

Additional Editor Comments (optional):

Reviewers' comments:

Reviewer's Responses to Questions

**Comments to the Author**

Reviewer #2: All comments have been addressed

2. Is the manuscript technically sound, and do the data support the conclusions?

Reviewer #2: Yes

3. Has the statistical analysis been performed appropriately and rigorously?

Reviewer #2: Yes

4. Have the authors made all data underlying the findings in their manuscript fully available?

Reviewer #2: Yes

5. Is the manuscript presented in an intelligible fashion and written in standard English?

Reviewer #2: Yes

Reviewer #2: (No Response)

**Do you want your identity to be public for this peer review?** For information about this choice, including consent withdrawal, please see our Privacy Policy

Reviewer #2: No

---

## [Editor Report · Acceptance letter]

PONE-D-24-57058R1

PLOS ONE

Dear Dr. Dhakad,

I'm pleased to inform you that your manuscript has been deemed suitable for publication in PLOS ONE. Congratulations! Your manuscript is now being handed over to our production team.

Kind regards,

on behalf of

Taimoor Hassan Farooq

Academic Editor

PLOS ONE